# Insights into Patient Heterogeneity in Sjögren’s Disease

**DOI:** 10.3390/ijms26136367

**Published:** 2025-07-02

**Authors:** Lisa Pecorelli, Kerstin Klein

**Affiliations:** 1Department of Rheumatology and Immunology, Inselspital, Bern University Hospital, University of Bern, 3008 Bern, Switzerland; 2Department for BioMedical Research, University of Bern, 3008 Bern, Switzerland

**Keywords:** Sjögren’s disease, heterogeneity, stratification, subtypes

## Abstract

Sjögren’s disease is multi-system autoimmune disease characterized by dryness of mucosal surfaces, fatigue, and pain. Heterogeneity among patients is a major obstacle for timely diagnosis, management of patients, and clinical trial design. Strategies for patient stratification are therefore desperately needed. In this review, we aimed to summarize current stratification approaches. Two major approaches for patient stratification are currently used. The first one is based on patient-reported symptoms and the subsequent analysis of the clinical and biological characteristics defining the identified clusters. The second strategy is based on the molecular stratification of patients, followed by the analysis of clinical characteristics along with other biological data. The combination of different approaches holds great potential to improve the recognition of patient subgroups and the development of tailored therapies. The current literature suggests that three to four subgroups of patients with SjD exist. Whether these subgroups represent disease stages or disease endotypes is still a matter of debate and will be a topic of future research.

## 1. Introduction

Sjögren’s disease (SjD) is a multi-system autoimmune disease characterized by sicca symptoms such as xerostomia and xeropthalmia. Clinical manifestations for SjD show a broad spectrum ranging from benign sicca symptoms to severe extraglandular manifestations, which present in 20 to 60% of patients [1,2]. Fatigue and pain are common and have a great impact on patients’ quality of life [3]. Furthermore, 5 to 10% of patients develop lymphoma [1], which is the leading cause of death [4].

The annual incidence and prevalence rates for SjD are estimated to range from 0.3 to 26.1 per 100,000 persons and 22 to 770 per 100,000 persons, respectively [5]. The variability of epidemiological data among studies might be caused by the repeated introduction of revised classification criteria for SjD and their potential misinterpretation, the misclassification of patients, and the potential introduction of confounding selection bias [6].

The current classification criteria for SjD consist of two major criteria, with the requirement of at least one being present, next to objective measurements of xerostomia and/or xeropthalmia [7]. One major criterion is the presence of anti-SSA/Ro autoantibodies [7], which are directed against cellular constituents of salivary gland epithelial cells that have undergone apoptosis [8]. The second major criterion is a focus score of ≥1/4 mm^2^, assessed by the histopathological evaluation of lymphocyte infiltrates in labial salivary gland biopsies [7].

Until recently, SjD was divided into “primary SjD” and “secondary SjD” (patients with another autoimmune disease) [9]. However, members of the International Sjögren International Collaborative Clinical Alliance Research Group suggested this categorization be dissolved and to substitute the term “secondary” in favor of the term “associated” SjD [2]. This suggestion emerged through the lack of evidence of a distinct pathophysiology between these two subgroups [2,9] and missing data regarding secondary SjD due to frequent exclusion in clinical trials [9].

## 2. The Need for Patient Stratification in Sjögren’s Disease

Substantial heterogeneity among patients with SjD is one of the major culprits for the existence of several unmet needs among patients and clinicians. Biomarkers that support diagnosis, particularly at early stages of the disease, and biomarkers that predict disease outcomes are desperately needed. Therapies for SjD are often limited to symptomatic approaches [5,10] or are still being evaluated in clinical trials [11]. This molecular heterogeneity contributes to gaps in knowledge regarding the pathobiology underlying SjD, particularly in different patient subgroups. To overcome these obstacles, different strategies for patient stratification have been developed, including approaches based on patient-reported symptoms, often followed by whole blood transcriptome or serum proteome analysis, the analysis of additional biological and clinical variables, and multi-omics profiling of whole blood samples as a basis for molecular pattern-driven stratification. These efforts have led to the identification of distinct subtypes of SjD, may facilitate patient recognition, and may homogenize patient subgroups in clinical trials aiming to develop tailored therapies in the future (Figure 1). The objective of this review was to provide an overview of different stratification approaches for patients with SjD, highlighting overlaps and disparities among studies (Table 1).

## 3. Interferon Signatures in Patients with SjD

The most widely expressed gene expression signature observed in patients with SjD reflects the activation of interferon (IFN) pathways and an increased expression of IFN target genes, commonly referred to as the IFN signature, which can be observed in periphery and salivary gland tissues in more than half of patients [18,19,20,21,22,23,24,25,26]. Further evidence of the role of IFN signaling pathways in SjD stems from genetic and epigenetic studies [27,28,29].

IFNs are a large group of cytokines that can broadly be divided into type I (IFN-α, IFN-β, IFN-ε, IFN-κ, IFN-ω), type II (IFN-γ), and type III (IFN-λ1, IFN-λ2, IFN-λ3) IFNs. Originally, IFNs were described to protect from viral infections, but they additionally modulate the adaptive immune response and are associated with several rheumatic autoimmune diseases [30,31]. Different types of IFNs act on different receptors but share the same downstream signaling, including components of the Janus-activated kinase/signal transducer activation of transcription (JAK/STAT) pathway [31]. Whereas type II IFNs are mainly produced by activated T cells, natural killer (NK) cells, and natural killer T (NKT) cells, type I and III IFNs are mostly produced by nucleated cells, with plasmacytoid dendritic cells (pDCs) being the most potent producers [25,32]. Increased levels of pDCs have been detected in minor salivary gland tissues of patients with SjD [18]. Type I and type II IFNs have been associated with SjD; however, there is also evidence for the potential role of type III IFNs in SjD [25].

An analysis of IFN signatures in the minor salivary gland tissues of patients with SjD revealed type I, type II, and mixed (type I + II) IFN signatures, with type II signatures being associated with higher focus scores [24]. Another study confirmed a stronger correlation of type II IFN signatures with focus scores in minor salivary gland tissues compared to type I IFN signatures [26]. A potential stratification of patients with SjD based on IFN signatures has already been suggested by Bodewes and coworkers in 2018 upon an analysis of two independent European SjD cohorts, identifying IFN-negative, type I IFN-, and type I plus type II IFN-positive patient subgroups [22]. Additionally, strong IFN signatures, assessed by whole blood transcriptome analysis in four independent cohorts of patients, have been identified in half of patients with SjD. These are driven by IFN-α, and not IFN-γ, and are associated with an increased frequency of systemic complications. The levels of IFN-α in serum strongly correlated with positivity for anti-SSA/Ro, anti- SSB/La, and rheumatoid factor [33], in line with findings by Tarn et al. indicating an association of high titers of anti-SSA/Ro and anti-SSB/La autoantibodies in patient clusters positive for IFN signatures [13].

Elevated IFN signatures in a subset of patients with SjD suggest that drugs targeting IFN pathways are at the forefront of therapeutic strategies and might have significant benefits in these patients. Among the therapeutic options are inhibitors of the JAK/STAT pathway, histone deacetylases (HDAC), phosphoinositide 3-kinase (PI3K), and blockade of interferon-α/β receptor (IFNAR) [34]. A transcriptome analysis of peripheral blood samples from patients with SjD, aiming to identify drugs for repurposing in SjD, provided additional evidence for targeting IFN pathways in SjD and identified HDAC and PI3K inhibitors as the most promising candidates [35].

## 4. Symptom-Based Stratification Approaches

### 4.1. The Newcastle Sjogren’s Stratification Tool

Given the poor correlation between objective measures and subjective symptoms reported by patients with SjD [36], the stratification of patients based on their symptoms may enable us to understand the entire pathophysiology related to SjD. In 2019, Tarn et al. [13] integrated five common patient-reported symptoms—pain, fatigue, dryness, anxiety, and depression, into the Newcastle Sjogren’s Stratification Tool, leading to the identification of four clusters in the UK Primary Sjögren’s Syndrome Registry (UKPSSR) cohort and their validation in two cohorts. With this approach, the authors revised the conventional categorization of patients into two groups based on the presence and absence of extraglandular manifestations. The four clusters were defined as “low symptom burden” (LSB), “high symptom burden” (HSB), “dryness dominant with fatigue” (DDF), and “pain dominant with fatigue” (PDF). The DDF cluster was of particular interest, given its association with the highest prevalence of lymphoma and increased levels of C-X-C motif chemokine ligand 13 (CXCL13), b2-microglobulin, and kappa free light chain (κ-FLC) [13]. High levels of CXCL13 have previously been identified to correlate with histomorphological parameters in salivary gland tissues, particularly focus scores and the presence of germinal centers, thus suggesting CXCL13 as a biomarker for histological involvement in SjD [37]. Transcriptome analysis of whole blood samples in the four patient clusters suggested differences in modular profiles, particularly regarding IFN and T cell modular activities. IFN-related molecular features were elevated in the DDF and LSB clusters, with predominance in the latter one [13].

To identify potential drivers and pathways that shape clinical manifestations identified in patients stratified using the Newcastle Sjogren’s Stratification Tool, a serum proteome analysis was performed, including “inflammation”, “immune response”, “organ damage”, “cardiovascular”, and “metabolism” proteomic panels. The DDF cluster was characterized by B cell hyperactivity, the highest prevalence of B cell lymphoma, and the presence of B cell-stimulating cytokines [38]. In line with the previous characterization of the DDF cluster [13], IFN activity scores, IFN-γ-induced chemokines, and chemokines associated with ectopic lymphoid structures were enriched. Whereas proteins in the DDF subtype pointed to glandular dysfunction, serum proteins enriched in the HSB and PDF clusters were inflammatory mediators and associated with antioxidant and DNA damage responses, as well as an altered energy metabolism [38].

Similarly to the Newcastle Sjogren’s Stratification Tool [13], Lee et al. [39] performed a longitudinal analysis of symptom-based clustering in a Korean cohort of patients with SjD, taking the symptoms pain, fatigue, dryness, anxiety, and depression into account. The authors identified three subgroups, including patients with “dryness dominant”, “high symptom burden”, and “low symptom burden” phenotypes [39], sharing similarities with clusters identified by Tarn et al. [13]. The autoantibody profiles did not differ between these clusters. Patients in cluster 2 (“dryness dominant”) were characterized by highest scores for dryness and fatigue, thus defining the cluster as the “high symptom burden” cluster [39], taking the patients’ perspectives into account [40]. Furthermore, this cluster was characterized by the highest frequency of extraglandular manifestations, such as arthralgia/arthritis, cutaneous involvement, peripheral neuropathy, and the highest ESSDAI scores. The identified clusters were temporally stable during a follow-up period of five years [39].

### 4.2. Therapeutic Significance of Patient Stratification Based on Patient-Reported Symptoms

The therapeutic significance of patient stratification based on patient-reported symptoms has been underscored by the re-evaluation of data from the JOQUER [41] and TRACTISS [42] clinical trials. Although the initial results from these trials were disappointing, the therapeutic efficacy of hydroxychloroquine and rituximab, respectively, differed between symptom-based stratified subgroups. In this context, patients from the DDF cluster had higher values of unstimulated salivary flow (USF) and stimulated salivary flow in comparison to the placebo group at week 48 of follow-up on rituximab treatment (TRACTISS trial). Additionally, the HSB cluster showed a reduction in ESSPRI scores compared to the placebo group upon hydroxychloroquine treatment (JOQUER trial) [13].

### 4.3. Patient Stratification Based on the Patient-Reported Symptoms Dryness, Fatigue, and Pain

One criticism of the above stratification approaches was that depression and anxiety were included in the stratification tools. A study conducted by McCoy et al. [36] in 2022 was based on the sole inclusion of SjD’s cardinal symptoms—dryness, pain, and fatigue—when using data from the Sjögren’s International Collaborative Clinical Alliance (SICCA) Registry and the Sjögren’s Foundation survey. Four clusters of patients were identified and characterized by “low symptom burden in all categories” (LSB), “dry with low pain and low fatigue” (DLP), “dry with high pain and low to moderate fatigue” (DHP), and “high symptom frequency/severity burden in all categories” (HSB). The patients included in the HSB cluster had the highest frequency of SjD-related symptoms, depression, and impaired quality of life (QoL). Furthermore, these patients had the highest rates of using opioid analgesics and non-prescription and prescription eye drops. The DLP cluster included patients with the highest positivity rates for rheumatoid factor and combined anti-SSA/Ro and anti-SSB/La autoantibodies, whereas the LSB cluster was characterized by the highest positivity rates for anti-SSA/Ro antibodies alone [36]. This suggests that the presence of anti-SSB/La autoantibodies may have additional value in terms of patient stratification while not being independently associated with SjD in the absence of anti-SSA/Ro autoantibodies and therefore not being part of the classification criteria [7]. Also, the DLP cluster included the highest proportion of patients with abnormal USF, a high ocular staining score (OSS), and a positive Schirmer’s test result. Although this cluster had a high symptom burden with regard to dryness, the authors classified this cluster as “low symptom burden” with the justification that only one of the three hallmarks of the disease (dryness) was increased, while the other two (pain and fatigue) were low. Interestingly, the LSB cluster was suggested to potentially represent an early stage of the DLP cluster, a hypothesis that was reinforced by the observation of patients in the former cluster being younger, on one hand, and the finding that patients in the DLP cluster showed a higher frequency of anti-SSA/Ro and anti-SSB/La autoantibody positivity, which could be a consequence of epitope spreading [36]. The hypothesis that one cluster represents an earlier disease stage of another cluster is still a matter of debate among different research teams [15,16,17,36,43].

### 4.4. Cluster Analysis Based on Patient-Reported Symptoms and Clinical and Biological Manifestations

In a large study by Nguyen et al. [12] in 2024, including 534 patients from the Paris-Saclay cohort and 395 patients from the ASSESS (Assessment of Systemic Signs and Evolution of Sjögren’s syndrome) cohort, 26 variables of patient-reported symptoms, including dryness, fatigue, and pain, and additional clinical and biological manifestations were analyzed. Three distinct subgroups of patients were identified: “B-cell active disease with low symptom burden” (BALS), “high systemic disease activity” (HSA), and “low systemic disease activity with high symptom burden” (LSAHS). The BALS cluster included patients with low symptom burden (assessed by low ESSPRI scores) and low systemic disease activity (assessed by low ESSDAI scores). Furthermore, these patients showed the highest positivity rates for rheumatoid factor and anti-SSA/Ro and anti-SSB/La autoantibodies. Patients from the HSA cluster suffered from a high symptom burden of dryness and fatigue and a high systemic disease activity. The third cluster, LSAHS, comprised patients with a high symptom burden related to dryness, fatigue, and pain, but low systemic disease activity. Lymphoma occurrences differed between the two cohorts considering that the French cohort showed lymphoma development in all clusters with a predominance in the HSA cluster, whereas in the ASSESS cohort, patients developed lymphoma in the BALS and HSA clusters but not in the LSAHS cluster [12].

In a follow-up study, these clusters were further evaluated by transcriptomic analysis for IFN scores in whole blood samples and an assessment of the serum levels of twelve cytokines and chemokines, leading to the discovery of potential biomarkers for these subgroups. Both the BALS and HSA clusters showed increased levels of IL-7, TNF-RII, and CXCL13. In addition, the HSA cluster was characterized by higher levels of FLT-3 and b2-microglobulin. In comparison to the other clusters, the BALS cluster was characterized by a high IFN signature, which was mainly driven by type I IFN. Given that the patients in the BALS cluster were diagnosed at a younger mean age, the potential role of IFN in early disease evolution was suggested by the authors. In addition, lymphoma occurred later compared to the HSA cluster; hence, the BALS cluster might represent a “pre-HSA” cluster [43]. A similar hypothesis was brought up by McCoy et al.’s team, who suggested that the LSB (“low symptom burden”) cluster might represent an early stage of the DLP (“dry low pain”) cluster due to the younger mean age at diagnosis of the patients in the LSB cluster and the higher frequency of autoantibodies in the DLP cluster [36]. If this was proven to be true, at least some of the identified subtypes identified by stratification tools would therefore better represent disease stages than SjD endotypes. In this context, stratification tools and identified biomarkers for patient subgroups might help to recognize patients at early stages of the disease and initiate treatment before irreversible tissue damage occurs.

A recent cross-sectional observational study by Fang et al. [14], including more than a thousand patients with SjD, categorized patients into four clusters based on symptoms, autoantibody profiles, and histopathological findings of labial salivary gland biopsies. In a subset of patients, transcriptome analysis of peripheral blood samples was carried out. In line with most other studies, four groups of patients with SjD were identified. Cluster 1 (“low systemic activity”) was characterized by the lowest systemic involvement, also reflected by the lowest ESSDAI scores, and showed no hematologic or serologic manifestations. In contrast, cluster 2 (“inflammatory”) included patients suffering from joint pain, dryness, and pain, assessed by the levels of myalgia, suggesting a “high symptom burden”. The transcriptome analysis pointed to the role of neutrophils and monocytes and detected elevated levels of cytokines and chemokines. Cluster 3 (“high systemic activity, inflammation”) was characterized by the highest proportion of patients with systemic symptoms such as cutaneous or renal involvement and hematological changes but without joint involvement. Furthermore, patients had high levels of fatigue. Patients in this cluster presented with the highest positivity rates for anti-SSA/Ro, anti-SSB/La, ANA, and anti-Ro-52 antibodies. IFN modules were enriched in clusters 2 and 3 compared to clusters 1 and 4. The Highest ESSDAI scores were observed in cluster 4 (“high systemic activity, non-inflammation”), which included patients suffering from joint discomfort and hematological changes. Inflammatory gene expression modules were downregulated in this cluster, particularly in comparison to cluster 3, which showed the highest enrichment of these factors [14].

### 4.5. Patient Group Classification in Clinical Routine Practice

To investigate how patients in routine practice can be classified by their attending rheumatologists into patient subgroups, aiming to gather real-world data, patients were clustered into five distinct subgroups based on organ involvement, pain, and fatigue (mental and physical). Cluster 1 (“low burden”) was characterized by low organ involvement, low mental fatigue but high physical fatigue, and no pain. Cluster 2 (“low burden, articular”) contained patients presenting with articular involvement and some levels of physical fatigue and pain, while cluster 3 (“moderate burden, articular”) presented with high levels of pain and the highest frequency of fatigue. This cluster was of particular interest, as it pointed to the discordance between a high patient-reported symptom burden and low systemic organ involvement [44]. This contradiction between subjective symptoms and objective measures highlighted again the need for symptom-based stratification approaches and was previously taken as a justification for this kind of stratification strategy in general. Cluster 4 (“high burden, multi-organ”) showed high organ involvement and high frequencies of pain and fatigue. This cluster contained patients with the highest clinical burden, the most severe symptoms, and the lowest proportion of patients with an improvement in the disease. Furthermore, this cluster showed the lowest treatment satisfaction, according to both patients and their physicians, whereas the opposite was observed for cluster 1. Patients in cluster 5 (“moderate burden, multi-organ involvement) presented with high rates of organ involvement, high frequencies of physical fatigue, and the highest frequency of pain [44]. Given the lack of laboratory measurements in this study, underlying mechanisms that drive distinct patient clusters were not identified.

## 5. Molecular Stratification Approaches

### 5.1. Advantages of Molecular Stratification Approaches

In contrast to the above-mentioned studies, in which transcriptome profiles of peripheral blood samples were used to further characterize symptom-based and clinically defined patient clusters, this approach itself was used to identify patient subgroups in the following section. This strategy considers that the experience of symptoms is subjective among patients, potentially leading to incorrect clustering. Therefore, obtaining objective values underlying patient stratification is needed. In this context, blood-based biomarkers are of particular interest, due to the convenience of sample collection. However, the molecular profiling of blood samples might not reflect SjD-specific processes. This was impressively illustrated by a cross-sectional analysis of the whole blood transcriptome and methylome signatures of more than 900 patients across seven autoimmune diseases, including SjD, systemic lupus erythematosus, systemic sclerosis, rheumatoid arthritis, primary antiphospholipid syndrome, and mixed and undifferentiated connective tissue diseases, and 256 healthy individuals [16].

In contrast to blood sampling, salivary gland biopsy collection is invasive, may cause complications and discomfort, and might better reflect the glandular involvement of SjD. Nevertheless, the profiling of affected tissue specimens potentially provides insights into disease-specific mechanisms. In rheumatoid arthritis and systemic sclerosis, histological and molecular profiling of the primarily affected tissues, synovium, and skin, respectively, has revealed the existence of distinct disease subsets that should eventually enable better patient stratification and potentially predict responses to specific therapies [45,46,47,48]. Transcriptome analysis of labial and parotid tissue samples has provided evidence that patients cluster into distinct groups that correlate with positivity for anti-SSA/Ro antibodies and rheumatoid factor, as well as focus scores [49]. Besides inflammation, the analysis of fibrosis in salivary glands might provide additional insights into disease subtypes. Salivary gland fibrosis in SjD has been shown to be associated with focus scores, USF, and specific gene expression profiles [50,51]. In a retrospective analysis of SjD patients grouped into four clusters based on the absence and presence of anti-centromere and anti-SSA/Ro antibodies, ultrasound shear-wave elastography, a noninvasive tool used to quantify the degree of glandular elasticity and fibrosis, was sufficient to identify patients with distinct clinical manifestations [52]. How changes in ultrasound shear-wave elastography are associated with expression profiles in salivary glands, whole blood, or other biological parameters remains to be investigated.

### 5.2. Transcriptome Analysis of Peripheral Blood Samples

The molecular profiling of whole blood samples across seven autoimmune diseases, including SjD, revealed three disease-associated clusters, defined as “inflammatory”, “lymphoid”, and “IFN” clusters, and one “undefined” cluster, which might represent an inactive or remission state of disease. In a follow-up analysis of a subset of patients at months six and fourteen, respectively, 96% of patients remained in the same cluster at six months. At 14 months, 46% of patients had switched between relapse and remission, whereas the pathological clusters remained stable [16].

The “IFN” cluster was associated with severe symptoms, including functional kidney abnormalities, thrombosis, and nervous system involvement. In contrast, patients from the “lymphoid” cluster presented with less aggressive symptoms, such as abdominal pain, constipation, and diarrhea. Beside transcriptome differences, serological differences were detected between clusters, with elevated levels of anti-citrullinated peptide (ACPA) antibodies in the “lymphoid” and “undefined” clusters and elevated levels of anti-dsDNA, anti-SSA/Ro, and anti-SSB/La antibodies in the “IFN” cluster. The “inflammatory” cluster was characterized by elevated levels of MMP-8, IL-1RA, and CXCL13. Similarly, patients from the “IFN” cluster presented with higher levels of CXCL13 and IL-1RA, as well as increased levels of CXCL10, BAFF, and TNF. No specific molecular patterns were associated with the “undefined” cluster. Together, the data suggest that the seven autoimmune diseases are driven by three distinct molecular pathways, given that all seven autoimmune diseases could be stratified into these four subgroups, assuming that one cluster represented inactive disease. Therefore, autoimmune diseases may not be stratified according to their diagnosis but rather by the underlying molecular patterns, thus leading to three potential therapeutic approaches [16].

In line with the four groups of patients identified across different autoimmune diseases, a transcriptome analysis of 351 patients with SjD enrolled in a prospective multicenter clinical cohort identified four subgroups. This study identified IFN-α as the main driver of variability among clusters and demonstrated an association of IFN-α with HLA gene polymorphisms. High levels of baseline IFN-α in serum correlated with ESSDAI biological domains, particularly IgG and gamma globulins, and were associated with the development of systemic complications during a 5-year follow-up [33].

In a small transcriptome study of patients with SjD by James et al., three groups of patients were identified that were further characterized by the serum levels of thirty cytokines, chemokines, and soluble receptors. The low IFN and inflammation signatures characterizing patients in cluster 1 suggest that this subset of patients may not profit from immune-modifying therapies and might be excluded in following trials focusing on INF and inflammatory pathways. This cluster included the highest proportion of patients suffering from joint pain and fatigue, whereas the highest dryness rates were observed in cluster 2. Patients from cluster 2 showed a strong IFN and inflammation modular network and the highest serum levels of the IFN-induced mediators CXCL9, CXCL10, and CXCL13, as well as elevated levels of LIGHT and BAFF. Furthermore, patients assigned to cluster 2 possessed the highest titers of anti-SSA/Ro and anti-SSB/La autoantibodies [15], in line with other studies showing a link between IFN signatures and the levels of CXCL13 and autoantibodies [13,33,43]. Cluster 3 was characterized by an intermediate IFN signature and the lack of an inflammation modular network. Several chemokines were elevated in this cluster, including IL-1a, IL-21, IL-2RA, CCL4, sE-selectin, and CXCL13. Also, a moderate elevation of CXCL10 and CXCL9 was observed.

### 5.3. Multi-Omics Profiling of Whole Blood Samples

A study conducted by Soret et al. [17] in 2021 was based on a multi-omics strategy of patient stratification using whole blood samples from more than 300 patients enrolled in a European SjD cohort. Transcriptome analysis enabled the identification of four distinct patient clusters (C1–C4). Clusters C1, C3, and C4 showed high IFN signatures and were further distinguished by the composition of type I and II IFN gene enrichment. In this regard, C1 displayed the strongest IFN signature (both type I and type II gene enrichment), whereas cluster C3 had an intermediate IFN signature (mainly type I) and cluster 4 had a low IFN signature (mainly type II). Cluster C2 was characterized by low type I and type II IFN signatures.

This study identified 257 genes differing between the four clusters, which were associated with three gene expression modules, including “IFN”, “lymphoid”, and “inflammatory and myeloid” pathways. Cluster 1 was mainly associated with the “IFN” gene module, whereas genes in cluster 3 were enriched in “lymphoid” pathways and genes in cluster 4 were enriched in “inflammatory and myeloid” pathways. Healthy individuals mainly clustered together with patients in cluster 2, suggesting that cluster 2 might comprise patients in remission or with inactive disease [17], similarly to findings by Barturen et al. [16]. Cluster 4 (“inflammatory”) was characterized by inflammatory signatures such as increased levels of IL-6, IL-10, IL-15, and STAT3. This cluster showed the most severe clinical phenotype and the highest ESSDAI scores. Cluster 1 showed an enrichment of CXCL13, IL-6, and IL-1RA. All clusters showed increased levels of CXCL10, CXCL13, BAFF, and GDF15, including cluster 2, although this cluster was referred as “healthy-like”. In addition, serological differences between clusters were observed, such as higher levels of anti-SSA/Ro and anti-dsDNA antibodies and circulating kappa and lambda free light chains in clusters 1 and 3, as well as elevated levels of rheumatoid factor in cluster 1 and the presence of ACPA in cluster 4 [17].

## 6. Discussion

Patient stratification approaches for SjD can be broadly divided into two strategies, with clustering based on patient-reported symptoms and clustering based on molecular profiles. Afterwards, the approach that is not used for clustering is often used, along with additional clinical, biological, and molecular data analysis, to further characterize patient subgroups. Irrespective of the stratification approach used, a common finding among studies is the existence of three to four subgroups of patients with SjD and the existence of a “healthy-like” cluster, which potentially represents an inactive disease state.

All studies distinguished clusters with low and high symptom burdens, as well as patients with low and high systemic involvement. Additionally, all studies investigated the three cardinal symptoms of SjD—dryness, pain, and fatigue. The similarity in the assessed parameters among different studies enabled the identification of significant overlaps in patient subgroups, despite the fact that the analyses were grounded in various statistical methodologies. In this regard, the BALS cluster (identified by Nguyen et al. [12]) shared similarities with the LSB (identified by McCoy et al. [36]), “low symptom burden” (identified by Lee et al. [39]), and “low burden” (identified by Gairy et al. [44]) clusters, considering that those clusters were defined by a low symptom burden and low systemic involvement, as well as low levels of dryness, fatigue, and pain. Accordingly, the HSA cluster [12] exhibited parallels with the DHP and HSB clusters [36], as well as with the “high symptom burden”, “dryness dominant” [39], and “high burden, multi-organ” [44] clusters, considering that all of them were characterized by a high symptom burden based on either dryness, pain, or fatigue or several of these items. In this respect, a standardized definition of “high symptom burden” would be of great benefit for comparisons of future studies.

The symptom-based stratification approaches extended through transcriptome profiling by Tarn et al. [13] and by McCoy et al. [36] shared great similarities, considering that both identified four clusters, including “low symptom burden”, “high symptom burden” and two additional clusters characterized by the extent of dryness, fatigue, and pain.

However, some studies contradicted each other. In this context, Nguyen et al.’s team [12] concluded that their clusters showed poor correlation to those identified previously using patient-reported symptoms [13], as patients from the HSB, DDF, and PDF clusters were spread among molecularly defined clusters [12]. This suggests that distinct molecular pathways may lead to similar symptoms or that the subjective characteristic of patient-reported symptoms is only partially sufficient to reflect distinct molecular mechanisms underlying disease.

The analysis of chemokine and cytokine profiles in the serum of patient clusters further underscored the molecular heterogeneity of SjD and pointed to the phenomenon that multiple immune pathways are dysregulated in SjD. Among the most common cytokines and chemokines analyzed were CXCL10, CXCL13, b2-microglobulin, IL-7, IL-1RA/IL-2RA BAFF, TNF, TNF-RII, MMP-8, and sE-selectin. Of particular interest was CXCL13, whose levels were elevated in several clusters across different studies, including the clusters BALS, HSA [12], DDF [13], C2 and C3 [15], “Inflammatory”, “IFN” [16], and C1 [17], suggesting a link between clusters enriched for IFN signatures and the serum levels of CXCL13 [13,15,33,43]. In addition, this chemokine was discussed as a potential biomarker for salivary gland pathology since the levels of CXCL13 were associated with salivary gland infiltration, focus scores, and the presence of germinal centers [37]. Furthermore, high levels of CXCL13 in the serum samples of patients with SjD have been suggested by several studies as a risk factor for the development of lymphoma [43,53,54,55].

Patient clusters characterized by high IFN signatures were detected across different studies, with high signatures identified in BALS [12], LSB and DDF [13], C2 and C3 [14], C2 [15], “IFN” [16], and C1, C2 (only anti-SSA/Ro-positive patients), C3, and C4 [17]. High IFN signatures were associated with a high symptom burden and more sever phenotypes [13,14,15,16,17]. However, both the BALS [12] and LSB [13] clusters were characterized by high IFN signaling but low symptom burden, highlighting the discordance of subjective symptoms and objective measures on the one hand as well as the poor understanding of underlying disease mechanisms on the other hand.

The retrospective stratification of patients enrolled in clinical trials and the re-analysis of data have underscored the benefits of stratification strategies [13,56]. Treatment efficacy upon treatment with hydroxychloroquine and rituximab was different in patients stratified based on symptoms [13]. Furthermore, higher baseline levels of CXCL13, IL-22, IL17A, IL17F, and TNF were associated with response to rituximab [57]. Peripheral blood transcriptome modules representing lymphocytes and erythrocytes at baseline were shown to be associated with the therapeutic response to hydroxychloroquine and leflunomide combination therapy in a clinical trial [56], pointing to the usefulness of molecular stratification approaches.

Together, these different stratification approaches have provided tremendous new insights into the mechanisms underlying SjD and are an important step towards the development of tailored therapies for patient subgroups. The molecular profiles of patient clusters relate, in most studies, to peripheral blood samples. In terms of biomarker discovery for disease subgroups, this is a valid strategy. However, in terms of understanding disease mechanisms, integrating an analysis of tissue samples, though less accessible, might provide another layer of information, particularly with respect to the identification of therapeutic targets.

## Figures and Tables

**Figure 1 ijms-26-06367-f001:**
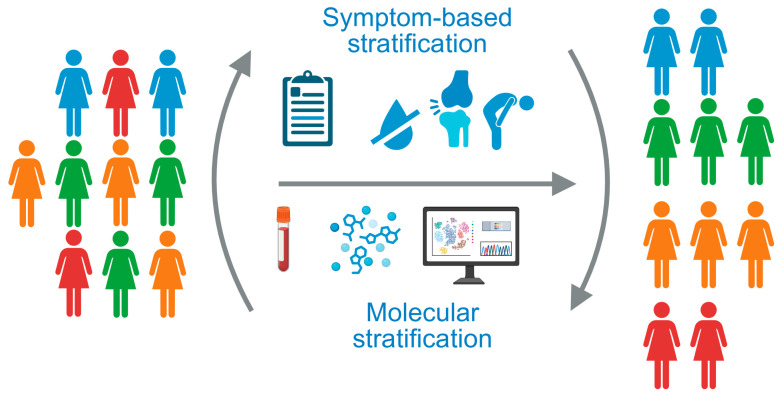
The heterogeneity of patients with Sjögren’s disease underlines the need for patient stratification strategies. Current approaches are based on patient-reported symptoms or molecular stratification of whole blood samples. Created in Biorender. Kerstin Klein. (2025) https://BioRender.com/uhwye8n.

**Table 1 ijms-26-06367-t001:** Overview of patient subgroups identified by different stratification approaches.

Reference	Characteristics	Cluster 1	Cluster 2	Cluster 3	Cluster 4	Cluster 5
Nguyen et al. [12]		**BALS** *B-cell-active disease with low symptom levels*	**HAS** *High systemic disease activity*	**LSAHS** *Low systemic activity; high symptom burden*	*-*	-
	**Clinical features**	Low ESSPRI, low ESSDAI, high frequency of lymphomas	High symptom burden of dryness, fatigue, and lower-body pain; high systemic disease activity	Hish ESSPRI, low ESSDAI, no lymphomas	-	-
	**Laboratory features**	High frequency of anti-SSA/Ro and anti-SSB/La antibodies and RF	Anti-RNP and anti-centromere antibodies	Lower frequency of anti-SSA/Ro and anti-SSB/La antibodies	-	-
	**Transcriptomic features**	IL-7, TNF-RII, CXCL13, and high IFN signatures	IL-7, TNF-RII, and CXCL13	Low IFN signatures	-	
Tarn et al. [13]		**LSB** *Low symptom burden*	**HSB** *High symptom burden*	**DDF** *Dryness dominant with fatigue*	**PDF** *Pain dominant with fatigue*	-
	**Clinical features**			Highest prevalence of lymphoma	-	-
	**Laboratory features**	Higher rates of anti-SSA/Ro and anti-SSB/La positivity		Higher rates of anti-SSA/Ro and anti-SSB/La positivity	-	-
	**Transcriptomic features**			CXCL13, b2-microglobulin		
Fang et al. [14]		**C1 “low systemic activity”**	**C2 “inflammatory”**	**C3 “high systemic activity, inflammation”**	**C4 “high systemic activity, non-inflammation”**	**Moderate burden, multi-organ involvement**
	**Clinical features**	Minimal systemic involvement, lowest ESSDAI scores	Serological changes, joint involvement, no systemic manifestations	High symptom burden, highest rate of fatigue	Joint and hematological involvement, highest ESSDAI scores, highest rates of dryness and fatigue	Articular and glandular involvement, high physical fatigue, highest pain
	**Laboratory features**	Lowest titers of autoantibodies		Highest positivity rates for ANA, SSA/Ro, and SSB/La antibodies	Highest rates of RF positivity	-
	**Transcriptomic features**	Decreased expression of modules associated with inflammation, monocytes, and T cells	Increased IFN signaling, elevated levels of cytokines and chemokines, overexpression of modules related to B and T cells, cellular growth, and metabolism	Increased IFN signaling, increased expression of modules associated with inflammation, monocytes, and T cells, downregulation of genes associated with neutrophil activation	Downregulation of most modules associated with inflammation, monocytes, and lymphocytes, overexpression of modules related to prostaglandins and cellular respiration	-
James et al. [15]		**C1**	**C2**	**C3**	*-*	
	**Clinical features**	Highest fatigue and joint pain, low dryness	Highest dryness, slightly higher ESSDAI scores	Low fatigue and joint pain	-	
	**Laboratory features**		Higher rates of anti-SSB/La positivity		-	
	**Transcriptomic features**	Low IFN signature, low inflammation	Highest levels of BAFF, CXCL10, and CXCL9, high levels of CXCL13	High levels of CXCL13 and CXCL10, highest levels of TNF-RII and sE-selectin, low inflammation	-	
Barturen et al. [16]		**Inflammatory**	**Lymphoid**	**Interferon**	**Undefined**	
	**Clinical features**	Fibrosis complications	Less aggressive phenotypes	Most severe phenotypes	Some clinical complications	
	**Laboratory features**	Elevated ACPA, anti-centromere B, and IgM anti-phosphorylcholine natural antibodies	Slightly enriched in ACPA, anti-centromere, and IgM anti-phosphorylcholine natural antibodies	Elevated levels of anti-dsDNA, anti-SSA/Ro, anti-SSB/La, and anti-U1-RNP antibodies		
	**Transcriptomic features**	Increased MMP-8, IL-1RA, and CXCL13		Increased CXCL10, BAFF, MCP2, TNF, IL1-RA, and CXCL13, association with HLA-class II genes	Nonspecific molecular patterns	
Soret et al. [17]		**C1 “interferon”**	**C2 “healthy like”)**	**C3 “intermediate IFN”**	**C4 “low IFN, inflammatory”**	
	**Clinical features**		Lowest ESSDAI scores, patients with glandular manifestations		Highest ESSDAI scores, patients with glandular manifestations	
	**Laboratory features**	Higher levels of ENA, anti-SSA/Ro, and anti-dsDNA antibodies, κ-FLC, and RF		Higher levels of ENA, anti-SSA/Ro, and anti-dsDNA antibodies and c κ-FLC	ACPA	
	**Transcriptomic features**	Highest IFN signature (type I + II), increased CXCL10, MCP2, TNFa, CXCL13, IL-6, IL-1RA, and BAFF	Weak type I + II IFN signatures, increased CXCL10, CXCL13, and BAFF compared to healthy controls	“Intermediate” high IFN signature (type I > type II), upregulation of IL-7-signaling, LXRL/RXR activation, higher levels of CXCL10, MCP2, TNFa, CXCL13, and BAFF	“Low” increase in IFN signatures (type II > type I), increased levels of CXCL10, CXCL13, and BAFF, downregulation of TGFb-associated modules	

## Data Availability

Not applicable.

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
