# Peer review of "Insights into Patient Heterogeneity in Sjögren’s Disease"

_ijms, 2025, doi:10.3390/ijms26136367_

Round 1
Reviewer 1 Report
Comments and Suggestions for Authors
In this review the authors discuss different approaches in an attempt to clarify the classification of Sjögren's syndrome patients. They report recent considerations regarding a clinical approach, based on the symptoms presented by the subjects and a biomolecular approach based, predominantly, on inflammatory indices.
The review is interesting, but difficult to consult. A reorganization is suggested that includes the subdivision into paragraphs and subparagraphs and the insertion of tables or diagrams or schemes that are lacking. Furthermore, an opinion is asked on the most recent discoveries that connect Sjögren's syndrome to fibrosis of the salivary glands or to their evolution towards salivary gland tumors. Could these indices also be used in the context of a possible classification?
In addition, I would like to ask if there is evidence of a similar stratification in other autoimmune diseases.
It would be interesting to evaluate if in these illustrated studies there are some limitations. Although clustering analysis is a useful exploratory tool, the methods are sensitive to the choice of clustering methods and a wide of symptoms revealed in this syndrome.
Author Response
Please find all comments in the uploaded word file.

Reviewer 2 Report
Comments and Suggestions for Authors
1. The English should be improved to enhance clarity and flow.
2. Some of the content in Table 1 could be simplified by focusing on key points, rather than presenting lengthy descriptions.
3. Since the number of core references discussed in the article is limited, including the authors' names and publication years for the key studies would help readers to follow the chronological development of this topic.
4. Interferons (IFNs) are mentioned multiple times, but this article lacks a detailed explanation of the classification and distinct roles of type I and type II IFNs, which would be beneficial for readers.
Author Response

(The authors gave the same response as above.)

Round 2
Reviewer 1 Report
Comments and Suggestions for Authors The manuscript has improved in the form and organization of paragraphs.